# Real-World Data on Immune-Checkpoint Inhibitors in Elderly Patients with Advanced Non-Small Cell Lung Cancer: A Retrospective Study

**DOI:** 10.3390/cancers17132194

**Published:** 2025-06-29

**Authors:** José del Corral-Morales, Carlos Ayala-de Miguel, Laura Quintana-Cortés, Adrián Sánchez-Vegas, Fuensanta Aranda-Bellido, Santiago González-Santiago, José Fuentes-Pradera, Pablo Ayala-de Miguel

**Affiliations:** 1Medical Oncology, Complejo Hospitalario Universitario de Cáceres, 10004 Cáceres, Spain; 2Medical Oncology, Hospital Universitario Virgen de Valme, 41014 Sevilla, Spain; 3Medical Oncology, Hospital Don Benito-Villanueva, 06400 Badajoz, Spain; 4Medical Oncology, Hospital de Mérida, 06800 Badajoz, Spain

**Keywords:** non-small cell lung cancer, immune-checkpoint inhibitors, elderly patients, immune-related adverse events, real-world data

## Abstract

Lung cancer represents the leading cause of cancer-related mortality, and the incidence of non-small cell lung cancer diagnosis in patients aged over 65 years is increasing. Despite this, older adults are often excluded from clinical trials evaluating immune-checkpoint inhibitors, leaving uncertainties about their effectiveness and safety in this population. This study aims to evaluate how age may influence the efficacy and safety of immunotherapy in patients with advanced non-small cell lung cancer. By analyzing data from 452 patients, the researchers also identify clinical characteristics that may serve as favorable prognostic factors. Our findings may contribute to building stronger evidence about using age as an exclusion criterion in clinical practice, especially when comprehensive geriatric assessment is not feasible.

## 1. Introduction

Lung cancer is the second most common malignancy worldwide and remains the leading cause of cancer-related mortality [1], with non-small cell lung cancer (NSCLC) accounting for approximately 85% of all cases [2]. The number of patients aged 65–74 years diagnosed with NSCLC is increasing globally, and this age group also shows the highest proportion of lung-cancer-related deaths [3].

In recent years, the treatment landscape of lung cancer has undergone a paradigm shift with the introduction of immune-checkpoint inhibitors (ICIs), particularly those targeting the programmed cell death protein 1 (PD-1) and its ligand PD-L1, which enhance anti-tumor immune responses. While chemotherapy has demonstrated efficacy and safety in elderly patients with a good Eastern Cooperative Oncology Group (ECOG) performance status [4,5], evidence regarding ICIs in this population remains limited. Although pivotal clinical trials included patients over 65 years, most were between 65 and 75, and it is uncertain whether this subgroup was homogeneous. Moreover, patients aged over 75 years have been underrepresented in registration studies [6,7,8,9,10,11,12,13,14,15,16]. Retrospective and single-arm studies suggest a similar safety profile across age groups but show conflicting results in terms of efficacy [17,18,19]. Therefore, there is a need for growing evidence to clarify whether elderly patients benefit equally from immunotherapy.

Treating elderly patients with NSCLC poses additional challenges for several reasons. Age-related physiological changes may lead to erroneous pharmacodynamic responses, which may impact drug metabolism, efficacy, and toxicity [20]. Additionally, immunosenescence—the age-associated decline in immune function—may influence the tumor progression and response to ICIs, although this remains a matter of ongoing debate [21,22].

Here, we present the results of a multicenter, retrospective study evaluating the real-world effectiveness and safety of ICIs in patients with advanced NSCLC, with a special focus on elderly subgroups.

## 2. Materials and Methods

### 2.1. Study Design and Participants

This retrospective, multicenter study included patients diagnosed with advanced or metastatic non-small cell lung cancer (NSCLC) who received treatment with PD-1 or PD-L1 immune-checkpoint inhibitors, including atezolizumab, nivolumab, pembrolizumab, or cemiplimab. These agents were administered either as monotherapy or in combination with platinum-based chemotherapy, according to clinical practice. Data were collected from four institutions in Extremadura and Andalucía (Spain) between April 2017 and December 2023. Patients who received fewer than two immunotherapy infusions were excluded. Patients were stratified into three age-based cohorts according to age at initiation of immunotherapy: ≤65 years (younger group, YG), 66–79 years (older group, OG), and ≥80 years (advanced older group, AOG). The data cutoff was 1 April 2024.

### 2.2. Objetcives

The primary objective was to assess the efficacy of ICIs in elderly patients (OG and AOG) compared with younger patients (YG) based on progression-free survival (PFS), overall survival (OS), and overall response rate (ORR). PFS and OS were defined as the time from the first immunotherapy cycle to radiologically confirmed disease progression (PD) or death from any cause. ORR included both complete (CR) and partial responses (PR). The disease control rate (DCR) included CR, PR, and stable disease (SD). Tumor burden was calculated as the sum of the longest diameters of up to five target lesions (maximum two per organ). Radiological response evaluations, including CR, PR, SD, and PD, were performed according to institutional criteria without a centralized review.

A secondary objective was to evaluate the safety profile across age groups, focusing on the incidence and severity of immune-related adverse events (irAEs), which were assessed according to the Common Terminology Criteria for Adverse Events (CTCAE) version 5.0.

### 2.3. Statistical Analysis

Patients who had no documented evidence of disease progression or death at the time of analysis were censored at the date of their last available follow-up. Continuous variables were dichotomized based on the median value, unless explicitly stated otherwise. Both PFS and OS were estimated using the Kaplan–Meier method, and differences in survival outcomes between groups were assessed using the log-rank test.

Hazard ratios (HRs) and 95% confidence intervals (CIs) for PFS and OS were calculated using univariable Cox proportional hazards models. Variables that demonstrated statistical significance in the univariable analysis, in addition to other clinically relevant variables—including pathological features, treatment-related parameters, and laboratory data—were subsequently incorporated into the multivariable Cox proportional hazards model to adjust for potential confounders. Comparisons were performed between the YG and both the OG and the AOG. A two-sided *p*-value ≤ 0.05 was considered statistically significant.

All statistical analyses were carried out using IBM SPSS Statistics, version 27.0.1 (IBM Corp., Armonk, NY, USA).

### 2.4. Review

A narrative literature review was performed by conducting a comprehensive search of the PubMed database, limited to publications in English and covering studies available up to January 2025. The search strategy included the following keywords: “efficacy,” “safety,” “elderly,” “metastatic non-small cell lung cancer,” “immunotherapy,” and “chemo-immunotherapy”. No exclusion criteria were applied. Articles were selected based on their relevance to the topic, and additional references were identified through manual screening of the cited literature.

## 3. Results

### 3.1. Patient Characteristics

A total of 452 patients with advanced NSCLC who initiated treatment with ICIs during the observation period were included, with a median age of 67 years (range: 40–89). Of these, 195 patients were ≤65 years (YG), 221 were aged 66–79 years (OG), and 36 were ≥80 years (AOG). In the overall cohort, 91.4% had a history of smoking (current or former smokers). The proportion of male patients was lower in the YG compared to the OG and AOG (74.4% vs. 88.2% and 86.1%, respectively). Adenocarcinoma histology was more prevalent in the YG than in the OG and AOG (67.2% vs. 53.4% and 41.7%, respectively), as was M1c disease according to the 8th edition of the TNM classification (48.7% in YG vs. 31.7% in OG and 22.2% in AOG) and the proportion of brain metastases (22.1% in YG vs. 12.7% in OG and 5.6% in AOG). PD-L1 expression ≥50% was less frequent in the YG and OG compared to the AOG (39.5% and 39.8% vs. 50%, respectively). Consequently, monotherapy with ICIs was more frequently administered in the AOG compared to the YG and OG (80.5% vs. 64.1% and 67.5%, respectively). Detailed baseline characteristics are summarized in Table 1.

### 3.2. Outcomes

PFS and OS in the entire cohort were 8.9 months and 13 months, respectively. No statistically significant differences in PFS were found between age groups: median PFS was 8.3 months in the YG, 8.4 months in the OG (HR = 0.98; *p* = 0.872 vs. YG), and 10.5 months in the AOG (HR = 0.89; *p* = 0.628 vs. YG) (Figure 1). Additionally, a non-significant trend toward longer OS was observed in the YG compared to the OG and AOG: median OS was 15.1 months in the YG, 10.3 months in the OG (HR = 1.23; *p* = 0.076 vs. YG), and 12.5 months in the AOG (HR 1.45; *p* = 0.070 vs. YG) (Figure 2).

Among patients with adenocarcinoma, no statistically significant differences in PFS were observed between groups: median PFS was 8.3 months in the YG, 10.2 months in the OG (HR = 0.91; *p* = 0.565 vs. YG), and 23.2 months in the AOG (HR = 0.70; *p* = 0.392 vs. YG). Similarly, no differences in OS were observed: median OS was 14.9 months in the YG, 13.2 months in the OG (HR = 1.10; *p* = 0.524 vs. YG), and 7.9 months in the AOG (HR = 1.40; *p* = 0.299 vs. YG). The follow-up of the AOG remains immature in this histology subgroup.

In the squamous cell carcinoma cohort, no statistically significant differences in PFS were found: median PFS was 6.8 months in the YG, 9.8 months in the OG (HR = 0.91; *p* = 0.701 vs. YG), and 9.2 months in the AOG (HR = 0.98; *p* = 0.943 vs. YG). No significant differences in OS were observed either: median OS was 15.6 months in the YG, 9.8 months in the OG (HR = 1.36; *p* = 0.167 vs. YG), and 15.4 months in the AOG (HR = 1.59; *p* = 0.137 vs. YG).

The DCR and ORR in the entire cohort were 57.1% and 36.5%, respectively. The DCR was similar across age groups: 56.9% in the YG, 57% in the OG, and 58.3% in the AOG; *p* = 0.964). Likewise, no significant differences in the ORR were observed between groups (40% in the YG, 33.5% in the OG, and 36.1% in the AOG; *p* = 0.387) (Figure 3). CRs were observed in 10 patients (5.1%) in the YG and in 10 patients (4.5%) in the OG, with none reported in the AOG. PRs were seen in 68 patients (34.9%) in the YG, 64 (28.9%) in the OG, and 13 (36.1%) in the AOG. The median treatment duration (MTD) of immunotherapy in the overall population was 4.6 months (range: 1.1–89.4). The MTD did not differ significantly between groups: 4.9 months in the YG, 4.6 months in the OG (HR = 1.00; *p* = 0.957 vs. YG), and 4.2 months in the AOG (HR = 0.94; *p* = 0.793 vs. YG).

### 3.3. Safety

irAEs of any grade were reported in 170 patients (37.6%). The median number of treatment cycles prior to the onset of the first irAE was 4 (range: 2–115). No statistically significant differences were observed in the incidence of irAEs across age groups: 42.8% in the YG, 32.6% in the OG, and 41.7% in the AOG (*p* = 0.106). Skin-related irAEs were the most prevalent in the YG (11.3%) and OG (9.5%), whereas gastrointestinal irAEs (diarrhea and colitis) were the most frequent in the AOG (13.9%). A detailed breakdown of irAE characteristics is presented in Table 1.

There were no significant differences in the incidence of grade ≥ 3 irAEs among the groups: 9.7% in the YG, 5.9% in the OG, and 8.3% in the AOG; *p* = 0.423). There were four treatment-related deaths: one 64-year-old patient due to dermatologic toxicity; two patients, aged 59 and 76, due to hepatotoxicity; and one 77-year-old patient due to pneumonitis. Treatment discontinuation occurred in 68.4% of patients due to disease progression or death, in 6.6% due to irAEs, in 5.1% following two years of sustained response, and in 6.4% for other reasons. Discontinuation due to irAEs was numerically more frequent in the AOG (11.1%) compared to the YG (5.6%) and OG (6.8%), although this difference did not reach statistical significance (*p* = 0.446). Discontinuation reasons by group are detailed in Table 2.

### 3.4. Prognostic Factors in the OG and AOG

A univariable and multivariable analysis was performed to identify potential prognostic factors for PFS and OS in the combined OG and AOG. These results are provided in Table 3. In the multivariable Cox regression model, squamous histology, ECOG PS 0-1, and the occurrence of irAEs were independently associated with significantly improved OS.

Within the OG and AOG, patients who experienced irAEs of any grade demonstrated significantly longer PFS compared to those who did not (median PFS: 18.1 vs. 6 months; HR = 0.48; *p* < 0.001) (Figure 4), irrespective of irAE severity. Similarly, patients in the YG with irAEs of any grade showed significantly longer PFS compared to those without irAEs (median PFS: 30.4 months vs. 4.3 months; HR = 0.28; *p* < 0.001).

Specifically in the OG and AOG cohorts, median PFS was not reached (NR) in patients with grade ≥3 irAEs, vs. 5.8 months in the no-irAE group (HR = 0.31; *p* = 0.006). Similarly, patients with grade 1–2 irAEs had a median PFS of 18.1 months compared to 5.8 months in those without irAEs (HR = 0.53; *p* < 0.001). A consistent trend was observed for OS. Patients who experienced irAEs had a significantly longer median OS of 29.8 months, compared to 8 months in those who without irAEs (HR = 0.45; *p* < 0.001). When stratified by severity, patients who experienced grade ≥3 and grade 1–2 irAEs had a median OS of 22.3 and 22.1 months, respectively, both notably higher than the median of 7.9 months observed in patients who did not experience any irAEs.

Additionally, PFS was significantly longer in patients who discontinued treatment due to irAEs compared with those who discontinued for other reasons or continued therapy. The latter group included patients who discontinued treatment due to disease progression, sustained response, or other causes, as well as those still receiving therapy. Median PFS was NR in the irAE-related discontinuation group vs. 7.4 months in the non-irAE discontinuation group (HR = 0.32; *p* = 0.012) (Figure 5). Moreover, patients who received corticosteroids specifically for the management of irAEs experienced significantly longer PFS compared to those who received corticosteroids during ICI therapy for other, non-irAE-related medical conditions. Median PFS was NR in the irAE indication group, whereas it was 13.1 months in the non-irAE indication group (HR = 0.49; *p* = 0.024). The median duration of corticosteroid treatment was 30 days (range: 5–150) in the irAE group and 12 days (range: 3–120) in the non-irAE group. The median dexamethasone-equivalent dose was 4 mg (range: 1–75) in the irAE group and 4 mg (range: 0.5–18.8) in the non-irAE group.

## 4. Discussion

This study shows no significant differences in PFS, OS, or ORR across three age groups in a large real-world cohort of patients with advanced NSCLC. Likewise, the incidence of irAEs was comparable between the younger, older, and advanced older groups. Importantly, the occurrence of irAEs was associated with prolonged PFS and OS, even in patients who discontinued treatment.

Globally, elderly patients have been defined as those aged ≥65 years, due to the age-related physiological changes and the presence of comorbidities in this age group [23]. However, given the heterogeneity within the elderly population, a threshold of 70 to 75 years is also considered acceptable, particularly when a comprehensive geriatric assessment is performed to guide oncologic treatment decisions [24,25].

Pivotal trials of atezolizumab, cemiplimab, nivolumab, and pembrolizumab included a considerable proportion of patients aged over 65 years [6,7,8,9,10,11,12,13,14,15,16,26]. The IMpower150, CheckMate-017, and -057 and KEYNOTE-010, -024, and -042 trials reported data on patients aged over 75 years, who accounted for approximately 10% of enrolled participants. A meta-analysis [27] of randomized controlled trials of ICIs demonstrated improved survival in patients aged between 65 and 70 years but not in those over 75 years. However, this analysis included patients with multiple tumor types, not limited to NSCLC. Similarly, Nosaki K et al. described a comparable efficacy and safety of pembrolizumab in patients over 75 years relative to the general population [18]. Despite these findings, several limitations affect the generalizability of the results: the studies were not specifically designed to compare outcomes across age groups, definitions of older populations varied widely, and analyses within these subgroups were often limited and inconclusive. Moreover, patients enrolled in clinical trials frequently differ from those seen in routine clinical practice, further limiting the applicability of these results to broader, real-world elderly populations.

The randomized phase III IPSOS trial compared first-line atezolizumab against single-agent chemotherapy in frail patients with an ECOG performance status of ≥2 or aged over 70 years. While the study was not restricted to elderly patients, it showed a survival benefit in patients aged under 80 years but not in those over 80 years [28]. Additionally, two other prospective trials found similar survival outcomes between elderly and general populations receiving nivolumab in a pretreated setting [29,30].

Few retrospective studies have assessed ICI outcomes by age, most of them reporting no major differences in efficacy across age groups, including patients aged over 80 years [17,19,31,32]. However, most of these studies were focused on ICI monotherapy. One exception is a retrospective single-arm study including 57 patients aged ≥75 years with advanced NSCLC treated with first-line nivolumab–ipilimumab reporting a median PFS of 7.1 months and a median OS of 14.1 months [33]. Notably, Lichtenstein et al. [19] reported shorter PFS and OS in octogenarians compared to younger patients. In our study, ICI—whether as monotherapy or in combination with chemotherapy—appeared to be as effective in elderly patients as in younger ones in terms of ORR and survival. In contrast to Lichtenstein’s series, we did not observe significant differences in PFS by age group. Of note, 42.8% of patients over 80 years in Lichtenstein’s cohort had ECOG ≥ 2, compared to 19.4% in ours. Moreover, the limited sample size of patients over 80 years in both studies (n = 28 and n = 36, respectively) may have influenced these results. Although a trend toward longer OS was seen in patients under 65 years in our cohort, this may reflect a lower competing risk of non-cancer mortality, higher rates of subsequent therapy, and a greater prevalence of favorable prognostic factors, such as adenocarcinoma histology, male sex, and chemo-immunotherapy regimen use. Additionally, the prevalence of M1c disease was significantly lower in both older cohorts, which may have influenced the lack of statistically significant differences in PFS and OS across age groups.

Prognostic factors such as ECOG ≥ 2, cachexia, and an elevated NLR have been previously associated with poorer outcomes in patients treated with immune ICIs. Although an NLR threshold of 5 has been proposed as a predictor of unfavorable outcomes with ICIs, the optimal cutoff remains controversial [34,35]. An NLR < 5 has been associated with improved OS in elderly patients, although this benefit has not consistently extended to PFS [17]. In our study, a low NLR was not associated with improved OS or PFS in the elderly cohort, which may be attributable to the use of a lower cutoff value (NLR < 3). Our multivariable analysis identified squamous histology, ECOG 0-1, and irAE occurrence as independent predictors of improved OS in elderly patients, with irAE development also being associated with longer PFS, providing novel insights into treatment stratification in this population.

In our series, as well as in previous retrospective studies [17,19,32], the rate of high-grade irAEs was similar across age groups. Despite the stricter inclusion criteria in clinical trials, the incidence of irAEs reported in those studies—up to 30%—[7,8,26] is comparable to our real-world findings. In line with Morinaga et al., who found similar rates of irAE-related treatment discontinuation between younger and older patients (19.3% vs. 24.8%, respectively) [17], we also found no significant differences across age groups (5.6%, 6.8% and 11.1% in the YG, OG, and AOG, respectively). The lower discontinuation rates in our cohort may reflect a better functional status, as a higher proportion of patients received ICI in the first-line setting (58.5% vs. 12.4% in Morinaga’s study). Furthermore, both studies demonstrated that discontinuation due to irAEs was associated with longer PFS, reinforcing the notion that irAEs may serve as favorable prognostic markers. Additionally, we provide detailed data on causes of treatment discontinuation, showing no significant differences across age groups and further supporting the safety profile of ICIs irrespective of patient age.

The impact of corticosteroids in patients receiving ICIs has been a subject of ongoing debate. Data from two large retrospective cohorts, conducted at Memorial Sloan Kettering Cancer Center and Gustave Roussy Cancer Center, indicated that baseline corticosteroid administration at doses exceeding 10 mg of prednisone or equivalent was associated with a reduced ORR, PFS, and OS [36]. Conversely, a large meta-analysis involving 4045 patients reported that corticosteroid use for the management of adverse events did not negatively impact survival outcomes, while corticosteroids prescribed for symptom control unrelated to irAEs were linked to worse prognosis [37]. In our analysis, patients receiving corticosteroids specifically for the management of irAEs had a longer median PFS compared to those who either did not receive corticosteroids or received them for other medical conditions unrelated to irAEs. These findings suggest that corticosteroid therapy, when administered appropriately for irAE management, does not compromise the efficacy of ICIs. Instead, poorer outcomes associated with corticosteroid use may reflect a worse underlying clinical condition in patients requiring steroids for other indications. Collectively, these results enhance our understanding of the prognostic implications of corticosteroid administration in the context of irAE management, both in the general population and specifically among elderly patients.

Our study has several limitations. First, the relatively small number of patients aged over 80 years and their selection for ICI monotherapy based on favorable characteristics may have introduced bias. Additionally, the retrospective design limited data collection on comorbidities and precluded a comprehensive geriatric assessment, which is essential for optimizing treatment decisions in older adults and ensuring an appropriate candidate selection. Nonetheless, the inclusion of patients who received chemo-immunotherapy indicates that some had a more aggressive disease, which may partially balance this selection bias. Despite these limitations, this study offers valuable real-world evidence on ICI outcomes in elderly patients with NSCLC.

## 5. Conclusions

This real-world, multicenter retrospective study suggests that ICIs provide comparable efficacy and safety in elderly patients with advanced NSCLC, including those aged over 80 years. The occurrence of irAEs and treatment discontinuation due to these events did not negatively impact PFS. On the contrary, the development of irAEs was associated with improved outcomes, suggesting potential prognostic value in older populations. Given the heterogeneity of this population, systematic geriatric assessment is essential for optimizing treatment selection and ensuring appropriate clinical decision-making. These findings reinforce the role of ICIs in elderly patients and emphasize the importance of individualized assessment rather than chronological age when guiding therapeutic strategies.

## Figures and Tables

**Figure 1 cancers-17-02194-f001:**
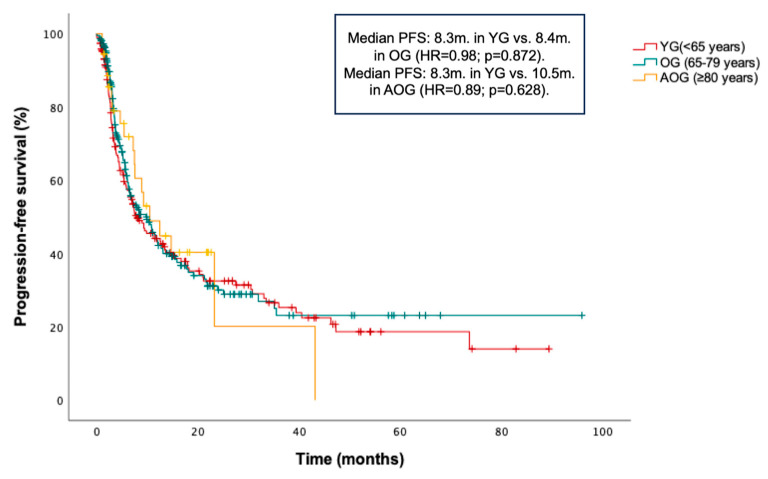
Kaplan–Meier curve for progression-free survival (PFS) according to age groups (≤65, 66–79, and ≥80 years).

**Figure 2 cancers-17-02194-f002:**
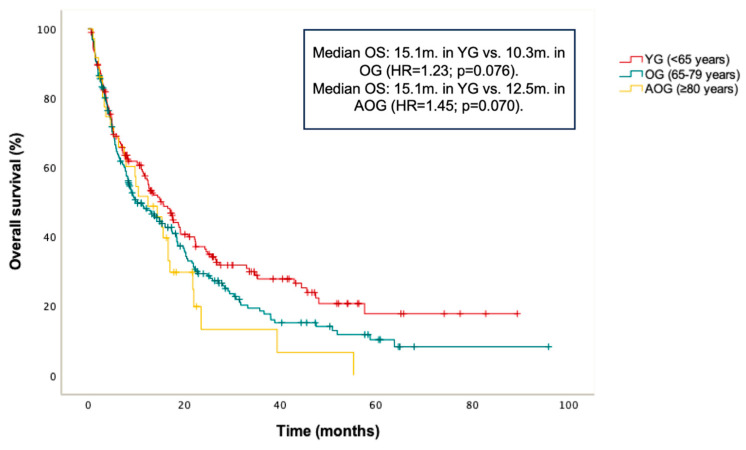
Kaplan–Meier curve for overall survival (OS) according to age groups (≤65, 66–79, and ≥80 years).

**Figure 3 cancers-17-02194-f003:**
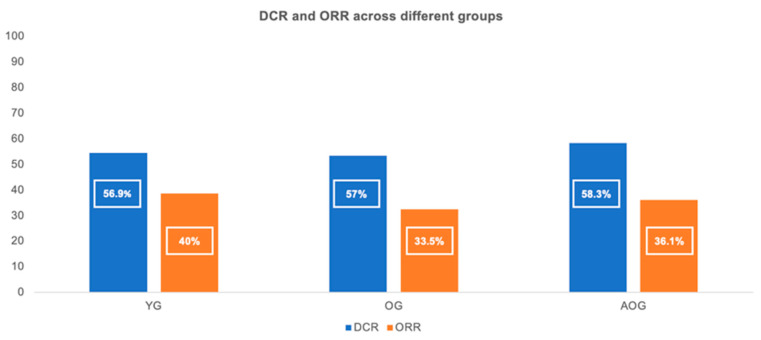
Disease control rate (DCR) and objective response rate (ORR) across the three age groups (≤65, 66–79, and ≥80 years).

**Figure 4 cancers-17-02194-f004:**
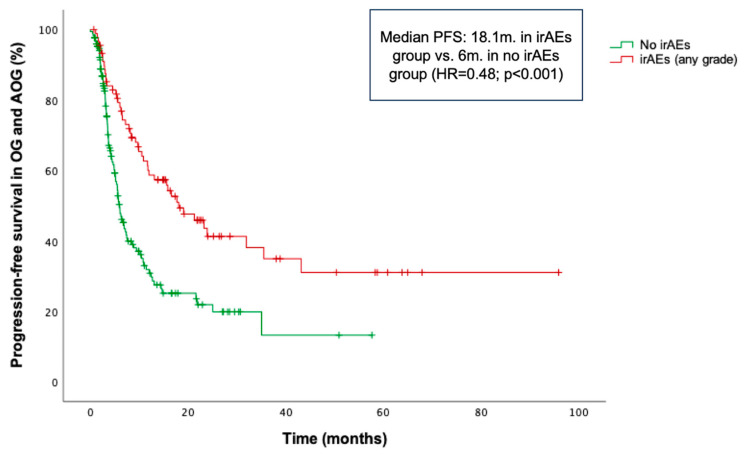
Kaplan–Meier curve for progression-free survival (PFS) according to the occurrence of immune-related adverse events (irAEs) in the combined older and advanced older groups (OG and AOG, ≥66 years).

**Figure 5 cancers-17-02194-f005:**
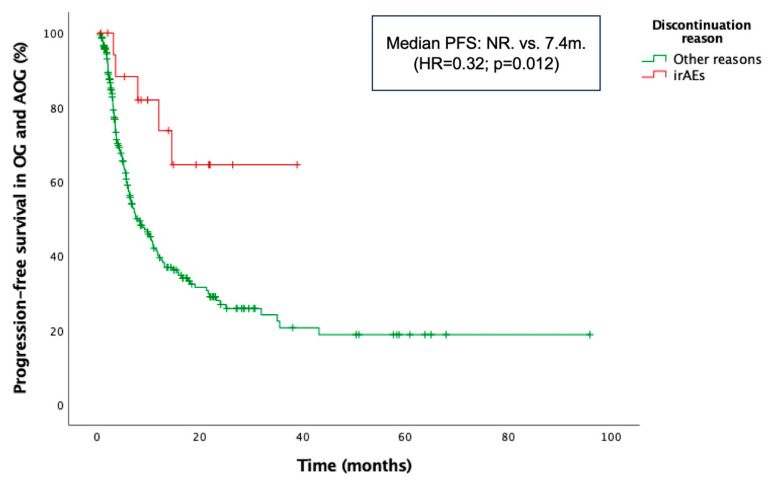
Kaplan–Meier curve for progression-free survival (PFS) according to the reason for treatment discontinuation in the combined older and advanced older groups (OG and AOG, ≥66 years).

**Table 1 cancers-17-02194-t001:** Baseline patient characteristics. Abbreviations: YG, younger group; OG, older group; AOG, advanced older group; ECOG PS, Eastern Cooperative Oncology Group Performance Status; Squamous cell carc., squamous cell carcinoma; NOS, not otherwise specified; ICI, immune-checkpoint inhibitor; irAEs, immune-related adverse events. Note: bold indicates statistical significance at *p* < 0.05.

	Overall, n (%) (n = 452)	YG n (%) (n = 195)	OG n (%) (n = 221)	AOG n (%) (n = 36)	*p* Value
**Sex**					***p* < 0.001**
Male	371 (82.1%)	145 (74.4%)	195 (88.2%)	31 (86.1%)	
Female	81 (17.9%)	50 (25.6%)	26 (11.8%)	5 (13.9%)	
**Smoking status**					***p* < 0.001**
Current smoker	204 (45.1%)	111 (56.9%)	84 (37.4%)	9 (25%)	
Never smoker	40 (8.8%)	12 (6.2%)	24 (10.8%)	4 (11.1%)	
Former smoker	208 (46.1%)	72 (36.9%)	113 (51.8%)	23 (63.9%)	
**ECOG PS**					*p* = 0.111
0–1	398 (88.1%)	173 (88.7%)	196 (88.7%)	29 (80.6%)	
≥2	54 (11.9%)	22 (11.3%)	25 (11.3%)	7 (19.4%)	
**Histology**					***p* = 0.009**
Adenocarcinoma	264 (58.4%)	131 (67.2%)	118 (53.4%)	15 (41.7%)	
Squamous cell carc.	152 (33.6%)	43 (22.1%)	90 (40.7%)	19 (52.8%)	
NOS	33 (7.3%)	19 (9.7%)	12 (5.4%)	2 (5.6%)	
Other	3 (0.7%)	2 (1.2%)	1 (0.5%)	0 (0%)	
**PDL1**					*p* = 0.224
≥50%	183 (40.5%)	77 (39.5%)	88 (39.8%)	18 (50%)	
1–49%	101 (22.3%)	41 (21%)	53 (24%)	7 (19.4%)	
<1%	101 (22.3%)	54 (27.7%)	42 (19%)	5 (13.9%)	
Unknown	67 (14.9%)	23 (11.8%)	38 (17.2%)	6 (16.7%)	
**Driver mutations**					*p* = 0.083
ALK	2 (0.4%)	1 (0,5%)	1 (0.5%)	0 (0%)	
EGFR	18 (4%)	12 (6%)	6 (3%)	0 (0%)	
KRAS	52 (10%)	29 (10%)	19 (9%)	3 (8.3%)	
BRAF	8 (2%)	4 (2%)	4 (2%)	0 (0%)	
MET ex14	2 (0.4%)	0 (0%)	1 (0.5%)	1 (2.8%)	
**Stage**					***p* = 0.024**
Recurrent IIIA	17 (3.8%)	8 (4.1%)	8 (3.6%)	1 (2.8%)	
Recurrent IIIB	17 (3.8%)	5 (2.6%)	10 (4.5%)	2 (5.6%)	
Recurrent IIIC	2 (0.4%)	2 (1%)	0 (0%)	0 (0%)	
M1a	158 (34.9%)	54 (27.7%)	86 (38.9%)	18 (50%)	
M1b	85 (18.8%)	31 (15.9%)	47 (21.3%)	7 (19.4%)	
M1c	173 (38.3%)	95 (48.7%)	70 (31.7%)	8 (22.2%)	
**M1 location**					
Brain	73 (16.2%)	43 (22.1%)	28 (12.7%)	2 (5.6%)	***p* = 0.007**
Liver	59 (13.1%)	26 (13.3%)	27 (12.2%)	6 (16.7%)	*p* = 0.752
Bone	115 (25.4%)	58 (29.7%)	51 (23.1%)	6 (16.7%)	*p* = 0.133
Adrenal	73 (16.2%)	35 (17.9%)	33 (14.9%)	5 (13.9%)	*p* = 0.660
**Treatment line**					*p* = 0.891
First line	264 (58.4%)	111 (56.9%)	130 (56.9%)	23 (63.9%)	
Second line	153 (33.8%)	66 (34.2%)	75 (33.9%)	12 (33.3%)	
Third and beyond	35 (7.8%)	18 (8.9%)	16 (7.2%)	1 (2.8%)	
**Treatment regimen**					*p* = 0.157
Chemotherapy-ICI	149 (32.9%)	70 (35.9%)	72 (32.5%)	7 (19.4%)	
Pembrolizumab	135 (29.9%)	52 (26.7%)	67 (34.4%)	16 (44.4%)	
Atezolizumab	116 (25.7%)	48 (24.6%)	56 (28.7%)	12 (33.3%)	
Nivolumab	52 (11.5%)	25 (12.8%)	26 (13.3%)	1 (2.8%)	
**IrAEs**					
Any grade	170 (37.6%)	83 (42.8%)	72 (32.6%)	15 (41.7%)	*p* = 0.106
Grade 3–4	35 (21%)	19 (9.7%)	13 (5.9%)	3 (8.3%)	*p* = 0.423
**Type of irAE**					
Gastrointestinal	23 (5.1%)	15 (7.7%)	3 (1.4%)	5 (13.9%)	***p* < 0.001**
Pneumonitis	21 (4.6%)	9 (4.6%)	9 (4.1%)	3 (8.3%)	***p* = 0.002**
Hepatotoxicity	18 (4%)	9 (4.6%)	9 (4.1%)	0 (0%)	*p* = 0.956
Skin toxicity	46 (10.2%)	22 (11.3%)	21 (9.5%)	3 (8.3%)	*p* = 0.427
Renal toxicity	8 (1.8%)	3 (1.5%)	5 (2.3%)	0 (0%)	*p* = 0.578
Thyroid dysfunction	25 (5.5%)	13 (6.7%)	10 (4.5%)	2 (5.6%)	*p* = 0.853
Hypophysitis	1 (0.2%)	1 (0.5%)	0 (0%)	0 (0%)	*p*= 0.522
**Subsequent line**					***p* = 0.045**
0	279 (61.7%)	108 (55.7%)	143 (64.7%)	28 (77.8%)	
1	134 (29.6%)	65 (33.5%)	61 (27.6%)	8 (22.2%)	
≥2	39 (8.6%)	22 (11.3%)	17 (7.7%)	0 (0%)	

**Table 2 cancers-17-02194-t002:** Reasons for treatment discontinuation by age group. Abbreviations: YG, younger group; OG, older group; AOG, advanced older group; DP, disease progression.

	Overall, n (%) (n = 452)	YG n (%) (n = 195)	OG n (%) (n = 221)	AOG n (%) (n = 36)	*p* Value
**Discontinuation reasons**					
DP or death	309 (68.4%)	134 (68.7%)	152 (66.8%)	23 (63.9%)	*0.769*
Toxicity	30 (6.6%)	11 (5.6%)	15 (6.8%)	4 (11.1%)	*0.446*
Maintained response	23 (5.1%)	13 (6.7%)	10 (4.5%)	0 (0%)	*0.200*
Other causes	29 (6.4%)	8 (4.1%)	18 (8.1%)	3 (8.3%)	*0.240*

**Table 3 cancers-17-02194-t003:** Multivariable analyses of PFS and OS in OG and AOG. Abbreviations: PFS, progression-free survival; OS, overall survival; YG, younger group; OG, older group; AOG, advanced older group; HR, hazard ratio; Squamous cell carc., squamous cell carcinoma; ECOG PS, Eastern Cooperative Oncology Group Performance Status; NLR, neutrophil-to-lymphocyte ratio; irAEs, immune-related adverse events. Note: bold indicates statistical significance at *p* < 0.05.

	PFS OG/AOG		OS OG/AOG	
	HR	*p*-Value	HR	*p*-Value
**Sex**				
Male				
Female	0.89 (0.48–1.68)	0.727	1.20 (0.72–2.00)	0.476
**Histology**				
Adenocarcinoma				
Squamous cell carc.	0.73 (0.47–1.13)	0.159	0.65 (0.45–0.94)	**0.022**
**ECOG**				
0–1				
≥2	1.81 (0.95–3.45)	0.070	2.56 (1.55–4.25)	**<0.001**
**NLR**				
<3				
≥3	0.96 (0.62–1.50)	0.857	1.33 (0.90–1.97)	0.155
**Stage**				
III/IVA				
IVB	1.15 (0.71–1.89)	0.570	1.20 (0.78–1.84)	0.407
**Brain metastases**				
No				
Yes	1.72 (0.90–3.28)	0.103	1.60 (0.91–2.80)	0.100
**PD-L1 (%)**				
<1%				
1–49%	0.76 (0.42–1.36)	0.352	0.88 (0.54–1.42)	0.598
≥50%	0.78 (0.44–1.39)	0.403	0.79 (0.48–1.30)	0.352
**Treatment line**				
First line				
Second line	1.31 (0.77–2.24)	0.320	1.14 (0.73–1.80)	0.563
Third and beyond	0.64 (0.19–2.09)	0.456	0.65 (0.23–1.81)	0.405
**irAEs**				
No irAEs				
irAEs	0.46 (0.29–0.73)	**0.001**	0.40 (0.26–0.60)	**<0.001**

## Data Availability

The original contributions presented in this study are included in the article. Further inquiries can be directed to the corresponding author.

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
