# Peer review of "Real-World Data on Immune-Checkpoint Inhibitors in Elderly Patients with Advanced Non-Small Cell Lung Cancer: A Retrospective Study"

_cancers, 2025, doi:10.3390/cancers17132194_

Round 1
Reviewer 1 Report
Comments and Suggestions for Authors
Thank you for allowing me to review this interesting review.
I found it very interesting to read.
Regarding the definition of elderly, in this retrospective study, elderly are defined as those aged 66 years or older. Globally, elderly are generally considered to be those aged 65 years or older; however, in some countries and regions, 70 or 75 years or older is considered to be elderly in clinical practice. Therefore, please add a discussion on the age cutoff of 70 and 75 years.
Additionally, the following reports on immune checkpoint inhibitors are important papers to consider, so please cite them.
Efficacy and safety of first-line nivolumab plus ipilimumab treatment in elderly patients (aged ≥ 75 years) with non-small cell lung cancer.
J Cancer Res Clin Oncol. 2025 Jan 23;151(1):43. doi: 10.1007/s00432-025-06089-x.
Author Response
Reviewer comment:
Regarding the definition of elderly, in this retrospective study, elderly are defined as those aged 66 years or older. Globally, elderly are generally considered to be those aged 65 years or older; however, in some countries and regions, 70 or 75 years or older is considered to be elderly in clinical practice. Therefore, please add a discussion on the age cutoff of 70 and 75 years.
Additionally, the following reports on immune checkpoint inhibitors are important papers to consider, so please cite them.
Efficacy and safety of first-line nivolumab plus ipilimumab treatment in elderly patients (aged ≥ 75 years) with non-small cell lung cancer. J Cancer Res Clin Oncol. 2025 Jan 23;151(1):43. doi: 10.1007/s00432-025-06089-x.
Response:
We appreciate this insightful comment and the suggested citation. We also recognize the importance of regional variability in cut-off age elderly definition. As suggested, we have incorporated relevant aspects from the cited work that may support the hypothesis of our study.
Reviewer 2 Report
Comments and Suggestions for Authors
This retrospective, multicenter study addresses the clinically important and timely question of the efficacy and safety of immune checkpoint inhibitors (ICIs) in elderly patients with non-small cell lung cancer (NSCLC), a population often underrepresented in pivotal trials. The authors present real-world data from a large cohort of 452 patients, with a key strength being the analysis of prognostic factors, particularly the correlation between immune-related adverse events (irAEs) and improved survival outcomes in the elderly.
Comments
1. Methodological Transparency and Adjustment for Geriatric Confounders
The authors correctly acknowledge that the study's "retrospective design limited data collection on comorbidities and precluded a comprehensive geriatric assessment". This is a critical limitation. While your multivariable model for the older cohorts identifies ECOG performance status as a key predictor, recent systematic work shows that more detailed Comprehensive Geriatric Assessment (CGA) domains—particularly frailty and functional status—are stronger independent predictors of immunotherapy tolerance and survival in older patients with advanced NSCLC. This leaves your findings susceptible to residual confounding by unmeasured factors like frailty, comorbidity burden, and treatment selection bias. (https://doi.org/10.1016/j.ctrv.2022.102394, doi: 10.3390/ijms26052120)
To strengthen the analysis, please consider:
- Retrospectively retrieving rapid geriatric metrics such as the G8 score or the Charlson Comorbidity Index (CCI) for the 257 patients in the OG and AOG cohorts. This data could be incorporated into multivariable Cox models or sensitivity analyses.
- Using propensity-score or inverse probability of treatment weighting (IPTW) to balance key prognostic variables that were significantly different at baseline, such as histology, stage, and the use of chemo-immunotherapy versus monotherapy across age strata.
2. Interpretation of irAE Findings and Disentangling Corticosteroid Exposure
Your observation that immune-related adverse events (irAEs) correlate with superior progression-free survival (PFS) and overall survival (OS) in older patients is striking and concordant with other datasets. (doi:10.1001/jamanetworkopen.2023.52302) However, your analysis of corticosteroid impact warrants a more nuanced approach. You report that "patients receiving corticosteroids specifically for the management of irAEs had a longer median PFS". This finding could be confounded by indication. The current model does not distinguish between:
i. Baseline corticosteroid use (for palliation or comorbidities), which consistently predicts inferior ICI outcomes. (doi: 10.3390/ijms231810292)
ii. High-dose steroids given to manage irAEs, which may not negatively impact the established anti-tumor response.
To clarify whether the survival advantage derives from the irAE itself (as a marker of a competent immune response) or from different patterns of steroid exposure, please consider stratifying the analysis by the timing and indication of steroid use, or including steroid exposure as a time-dependent covariate in the Cox model.
Author Response
Reviewer comment:
- Methodological Transparency and Adjustment for Geriatric Confounders:
The authors correctly acknowledge that the study's "retrospective design limited data collection on comorbidities and precluded a comprehensive geriatric assessment". This is a critical limitation. While your multivariable model for the older cohorts identifies ECOG performance status as a key predictor, recent systematic work shows that more detailed Comprehensive Geriatric Assessment (CGA) domains—particularly frailty and functional status—are stronger independent predictors of immunotherapy tolerance and survival in older patients with advanced NSCLC. This leaves your findings susceptible to residual confounding by unmeasured factors like frailty, comorbidity burden, and treatment selection bias. (https://doi.org/10.1016/j.ctrv.2022.102394, doi: 10.3390/ijms26052120)
To strengthen the analysis, please consider:
- Retrospectively retrieving rapid geriatric metrics such as the G8 score or the Charlson Comorbidity Index (CCI) for the 257 patients in the OG and AOG cohorts. This data could be incorporated into multivariable Cox models or sensitivity analyses.
- Using propensity-score or inverse probability of treatment weighting (IPTW) to balance key prognostic variables that were significantly different at baseline, such as histology, stage, and the use of chemo-immunotherapy versus monotherapy across age strata.
- Interpretation of irAE Findings and Disentangling Corticosteroid Exposure
Your observation that immune-related adverse events (irAEs) correlate with superior progression-free survival (PFS) and overall survival (OS) in older patients is striking and concordant with other datasets. (doi:10.1001/jamanetworkopen.2023.52302) However, your analysis of corticosteroid impact warrants a more nuanced approach. You report that "patients receiving corticosteroids specifically for the management of irAEs had a longer median PFS". This finding could be confounded by indication. The current model does not distinguish between:
- Baseline corticosteroid use (for palliation or comorbidities), which consistently predicts inferior ICI outcomes. (doi: 10.3390/ijms231810292)
- High-dose steroids given to manage irAEs, which may not negatively impact the established anti-tumor response.
To clarify whether the survival advantage derives from the irAE itself (as a marker of a competent immune response) or from different patterns of steroid exposure, please consider stratifying the analysis by the timing and indication of steroid use, or including steroid exposure as a time-dependent covariate in the Cox model.
Response:
We recognize that the lack of a comprehensive geriatric assessment is a major limitation of our study. However, retrospective reconstruction of tools such as the G8 score or the Charlson Comorbidity Index (CCI) is challenging in our cohort due to missing data and the subjective nature of some of the required items (e.g., food intake, self-perceived health status), which were not systematically recorded. In light of these limitations, we chose to use ECOG performance status which also reflects the eligibility criteria commonly used in pivotal clinical trials and clinical practice where a full geriatric evaluation is not available. Regarding IPTW analysis, we appreciate the suggestion but the limited time frame for submitting the revised manuscript prevents us from performing a reanalysis using advanced statistical weighting methods.
Regarding the comment on corticosteroid (CS) use, we have refined our analysis by comparing patients who received CS exclusively for the management of irAEs versus those who received them for other, non–irAE-related indications. We hope this distinction helps clarify that the observed survival advantage is more likely related to the occurrence of irAEs. Additionally, we now report the median dexamethasone-equivalent dose and the median duration of CS treatment. We recognize that the impact of corticosteroids on ICI efficacy remains a clinically relevant and debated issue; for this reason, we are currently preparing a separate manuscript specifically focused on this topic.
Reviewer 3 Report
Comments and Suggestions for Authors
The objective of this retrospective study was to compare the efficacy and safety of immune checkpoint therapy in elderly patients with advanced non-small cell lung cancer (NSCLC) with those in adult patients under age 65. The evaluation was conducted in a total of 452 NSCLC patients who have been treated with immune checkpoint inhibitors (ICIs). No statistically significant differences were observed in the median progression-free survival (PFS), median overall survival (OS) incidence of immune-related adverse events (reAEs) among the younger group (YG; under age 65), older group (OG; age 66-79) and advanced older group (AOG; over age 80). In patients at age 66+, those experiencing irAEs demonstrated significantly longer PFS and compared to those who did not experience irAEs, suggesting that irAEs may serve as a favorable prognostic factor in elderly patients.
Major comments:
The statistical significance found in NSCLC stages across the three age groups may influence the observed absence of statistically significant differences in PFS and OS. The authors should provide justification to minimize any potential statistical bias.
The impact of age on the efficacy of ICI therapy in patients with lung adenocarcinomas and squamous cell carcinomas should be evaluated separately.
The association between the incidence of irAEs and PFS should be assessed in patients under age 65 too.
Author Response
- Reviewer comment:
The statistical significance found in NSCLC stages across the three age groups may influence the observed absence of statistically significant differences in PFS and OS. The authors should provide justification to minimize any potential statistical bias.
The impact of age on the efficacy of ICI therapy in patients with lung adenocarcinomas and squamous cell carcinomas should be evaluated separately.
The association between the incidence of irAEs and PFS should be assessed in patients under age 65 too.
Response:
We would like to thank you for your comments. We fully agree that the differences found in NSCLC stages between groups may represent a potential bias contributing to the lack of statistically significant differences in PFS. Accordingly, we have added a sentence in the Discussion acknowledging this as a potential limitation that may help explain our findings.
Additionally, we have included statistical analyses of PFS and OS across the three age groups within the adenocarcinoma and squamous cell carcinoma subgroups, finding no statistically significant differences between age groups within each histological subtype. Furthermore, we have added data on the association between irAEs and PFS in the YG, which supports the findings observed in the older cohorts.
Round 2
Reviewer 1 Report
Comments and Suggestions for Authors
The authors appear to have responded appropriately to the reviewers' comments and we therefore consider the paper acceptable for publication.
Author Response
Dear reviewer,
Thank you again for your time and consideration.
Yours faithfully,
José del Corral-Morales
Reviewer 3 Report
Comments and Suggestions for Authors
The authors have addressed all of my concerns.
Author Response
Dear reviewer,
Thank you again for considering our article in Cancers.
Yours faithfully,
José del Corral-Morales